# EfficientSeg: An Efficient Semantic Segmentation Network

**Abstract.** Deep neural network training without pre-trained weights and few data is shown to need more training iterations. It is also known that, deeper models are more successful than their shallow counterparts for semantic segmentation task. Thus, we introduce EfficientSeg architecture, a modified and scalable version of U-Net, which can be efficiently trained despite its depth. We evaluated EfficientSeg architecture on Minicity dataset and outperformed U-Net baseline score (40% mIoU) using the same parameter count (51.5% mIoU). Our most successful model obtained 58.1% mIoU score on the official Minicity challenge.

**Keywords:** semantic segmentation, few data, MobileNet

## 1 Introduction

Typical machine learning approaches, especially deep learning, draw its strength from the usage of a high number of supervised examples[15]. However, reliance on large training sets restricts the applicability of deep learning solutions to various problems where high amounts of data may not be available. Thus, generally in few shot learning approaches, it is very common to start the network training using a pre-trained network or network backbone to obtain prior knowledge [24] from a larger dataset like ImageNet[5]. However, for the tasks defined on domains that are different from that of natural images such as for medical image segmentation [19, 13], it is not meaningful to start from pre-trained weights. This distinction makes learning from scratch using a low number of data instances, an important objective. This is also the objective of the newly emerging data-efficient deep learning field.

In [7], the authors argued that, non-pre-trained models can perform similar to their pre-trained counterparts even if it takes more iterations and/or fewer data to train. Also in [26], it is shown that, with stronger data augmentation the need to pre-train the network lessens. Even when using pre-trained networks, there is strong evidence that data augmentation improves the results [10, 17, 2].

In semantic segmentation, it is known that building deeper networks or using deeper backbones affects the results positively [8, 16]. Yet deeper networks come with limitations. Ideally, a baseline network which is subject to scaling should be memory and time-efficient. The latter is due to the fact that the number

of needed training iterations will be increased for a large network. Using Mo-
bileNetV3[9] blocks, we are able to create a baseline model which is still expres-
sive and deep with a lower parameter count. Regarding all these considerations,
in this article, we present a new deep learning architecture for segmentation, us-
ing MobileNetV3 blocks. As we focused on the problem of training with few data,
we evaluated our network in Minicity dataset[1], which is a subset of Cityscapes
[3].

## 2   Related Work

**Semantic Segmentation.** Computer vision problems focus on extracting use-
ful information from images automatically such as classifying objects, detecting
objects, estimating pose and so on. Semantic segmentation is one such prob-
lem where the main concern is to group the pixels on an image to state what
pixels belong to which entity in the image. Semantic segmentation finds many
applications in real life problems yet we can divide the efforts on the field into
two main categories: offline segmentation and real-time segmentation. Real-time
segmentation networks need to be both fast and accurate, with this constraint
they generally have lower mIoU compared to their counter-parts. To our knowl-
edge currently the state-of-the-art is U-HarDNet-70[1] with reported 75.9% class
mIoU and 53 frames per second with a 1080Ti GPU. On the other hand, offline
segmentation has no time concerns thus the proposed solutions are generally
slower. To our knowledge, the state of the art technique on offline Cityscapes
segmentation is HRNet-OCR[23] with a class mIoU of 85.1%. We next describe
the most popular architectural paradigm in image recognition, namely the Mo-
bileNet.

    **MobileNet Blocks.** With the increasing popularity of CNNs, the demand
on easy-to-access applications based on CNNs have also increased. One way
to establish the demanded accessibility is to use mobile devices, yet the com-
petition on image recognition challenges generally pushed CNN networks into
being too big to run on mobile devices. In this environment, there are two main
solutions to make mobile CNN applications feasible: running the networks in
powerful servers for external computation or using smaller networks to fit in
mobile devices. In this paper, we focus on the second solution, which aims at
creating smaller networks. Howard et al. introduced a family of networks called
MobileNets[10] with this motivation. The main idea behind MobileNets is uti-
lizing Depthwise Separable Convolutional (DSC) layers. DSC layer is very much
like a standard 2D convolutional layer and serves the same purpose yet it is both
smaller in number of parameters and faster compared to its counterpart. Figure 1
depicts the difference between a standard convolution layer and DSC layer. Mo-
bileNet architecture has two more improved versions namely MobileNetV2[20]
and MobileNetV3[9], before going into the details of MobileNetV3, we describe

---

[1] https://github.com/VIPriors/vipriors-challenges-toolkit/tree/master/semantic-
segmentation

MobileNetV2 and another work based on it, EfficientNet[22].

**MobileNetV2 Blocks and EfficientNet.** MobileNetV2 relies on two main components: depthwise separable convolutional layers and inverted residual architecture with linear bottlenecks. Inverted residual architecture is implemented by adding a middle phase called expansion phase, inside MobileNetV2 blocks the input tensor are expanded into having $t \times d$ depth with a convolution operation $t$ and $d$ are expansion ratio and depth of the input tensor respectively, after the expansion phase depthwise separable convolution phase follows. EfficientNets[22] are a family of networks which was built to be small, fast and accurate on image classification task. It consists of blocks pretty similar to MobileNetV2, yet instead of making the networks mobile, the authors used the advantages of MobileNetV2 blocks to create bigger networks, namely EfficientNets, are have significantly smaller number of parameters compared to their similar performing counterparts thus they are both memory and time efficient. After the success EfficientNet has achieved, Howard et al. published another work which is called MobileNetV3[9].

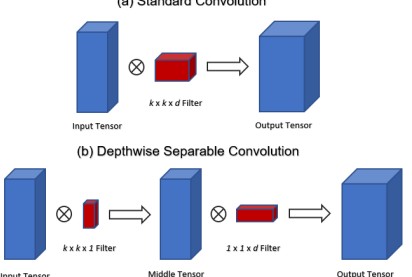

**Fig. 1.** Figure shows the difference between a standard convolution layer (a) and a depthwise separable convolution layer (b), depthwise separable layer consists of two convolution operations which decreases the number of parameters. In the figure "k" is the kernel size and "d" is the depth of the input tensor.

**MobileNetV3.** We use MobileNetV3 as the building blocks of our network EfficientSeg. Howard et al. added a Squeeze-and-Excite[11] operation to the residual layer and introduced a new architecture scheme. In our work we use this architecture to create a U-shaped semantic segmentation network. We will discuss further details in the following sections.

**Data augmentation.** As stated in Section 1, data augmentation is important for learning from few data. In traditional neural network training, transformations like flipping, cropping, scaling and rotating are highly used. In [18], [4]

and [12] more complex data augmentation methods like JPEG compression, local copying of segmentation masks, contrast, brightness and sharpness changes, blurring are suggested. There are also data augmentation methods focusing on generating new data by GANs or style transfer[25, 21, 6], but they are out of scope for the Minicity segmentation task since they are not generally applicable for training from scratch.

## 3   Method

In this paper, we present a new neural architecture called EfficientSeg, which can be counted as a modified version of the classic U-Net architecture[19] by alternating the blocks with inverted residual blocks which are presented in MobileNetV3[9].

The architecture of the EfficientSeg network, which is illustrated in Figure 2, is a U-shaped with 4 concatenation shortcuts, between an encoder and a decoder. Our encoder which is the down-sampling encoding branch of the network is like a MobileNetV3-Large classifier itself without the classification layers, whereas the decoder is its mirror symmetric version, where the down-sampling is replaced with upsampling operation. In the decoder part, we need to upsample the input tensors to retrieve a segmentation mask image which is the same size as the input image. We apply an upsample with bilinear interpolation and a scale factor 2 at each block where its symmetric is a downsample block on the encoder side.

We have 4 shortcut connections across from the encoder towards the decoder at the same layer. Each shortcut is done by concatenating the input of a down-sampling block in the encoder part with the corresponding upsampled output in the decoder part. In this way, we enable the network to capture the fine details through these shortcuts rather than solely preserving them in the bottleneck.

As in MobileNetV3 blocks, a width scaling parameter to upscale the network also exists in EfficientSeg, making it suitable to create networks of different scales. We will be discussing two of them which are EfficientSeg (1.5) which has the same number of parameters as baseline the U-Net in Minicity Challenge and also our larger network EfficientSeg (6.0).

## 4   Experiment

In our experiments, we train the EfficientSeg network with $384 \times 768$ sized cropped images using Adam[14] optimization algorithm with a learning rate of *lr=1e-3* at the start. We divide the learning rate by 10 at . As the objective function, we use a weighted cross-entropy loss. In the dataset, we observe that some of the categories are underrepresented relative to the others. We incorporate that information into the objective function in the form of increased weights: a weight of 2 (wall, fence, poll, rider, motorcycle, bicycle) and a weight of 3(bus, train, truck) are used for the rare classes. For every epoch, 20 extra images for each rare class are also fed to the network.

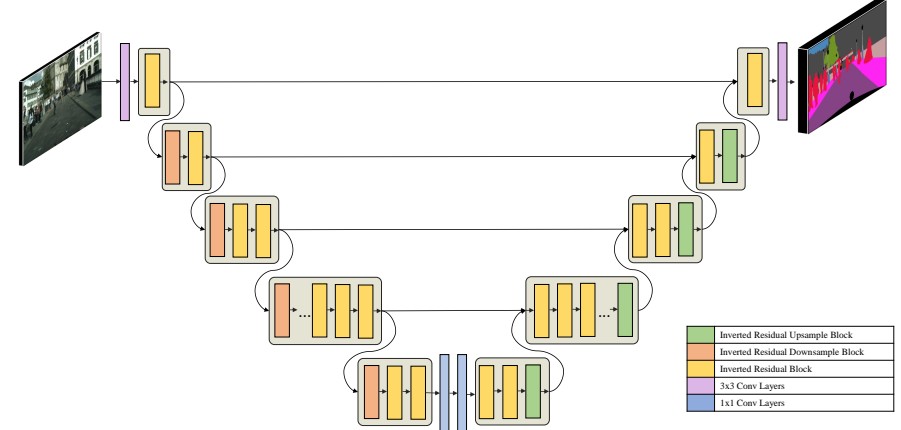

**Fig. 2.** EfficientSeg architecture. There are 5 different type of blocks. Inverted Residual Blocks are MobileNetV3 blocks described as in the paper. 1x1 and 3x3 blocks are standard convolution blocks which has activation and batch normalization. Downsampling operations are done with increasing the stride and for upsampling, linear interpolation is used.

Deciding on which data augmentations to use requires prior knowledge of the domain [4]. Since in our train set we have few objects of same category having different color and texture properties, we decided to reduce the texture dependency and increase the color invariance by (i) multiplying hue and brightness values of the image by uniformly distributed random values in $(0.4, 1.6)$, and (ii) JPEG compression. We also did (iii) non-uniform scaling, (iv) random rotation ($\pm 20°$) and (v) flipping as in standard deep learning approaches. At evaluation time, we feed the network with both the original test images and their flipped versions, then calculate average of their scores to obtain the final segmentation.

Utilizing nearly the same parameter count by using a depth parameter of 1.5, we obtain an mIoU score of 51.5% on the test set whereas baseline U-Net model has a score of 40%. To further improve the model we also tested with a depth parameter of 6.0 and obtain an improved mIoU result of 58.1%. To demonstrate the importance of texture based data augmentation, we also train the network without the aforementioned augmentations. As can be seen in Table 1, using both the aforementioned augmentation strategy and increasing the depth of the network, we obtain our highest score.

It is also worth mentioning that, the effect of the aforementioned data augmentation techniques, is more significant than depth up-scaling. This result empirically shows the importance of texture based data augmentation.

| | EfficientSeg (1.5) | EfficientSeg (6.0) w/o aug. | EfficientSeg (6.0) |
|---|---|---|---|
| road | 0.960 | 0.954 | 0.962 |
| sidewalk | 0.707 | 0.685 | 0.738 |
| building | 0.846 | 0.832 | 0.864 |
| wall | 0.277 | 0.165 | 0.318 |
| fence | 0.285 | 0.197 | 0.304 |
| pole | 0.449 | 0.471 | 0.517 |
| traffic light | 0.239 | 0.382 | 0.450 |
| traffic sign | 0.491 | 0.517 | 0.615 |
| vegetation | 0.885 | 0.888 | 0.899 |
| terrain | 0.501 | 0.464 | 0.576 |
| sky | 0.912 | 0.919 | 0.932 |
| person | 0.580 | 0.575 | 0.710 |
| rider | 0.222 | 0.179 | 0.353 |
| car | 0.864 | 0.842 | 0.899 |
| truck | 0.342 | 0.106 | 0.497 |
| bus | 0.264 | 0.128 | 0.325 |
| train | 0.169 | 0.002 | 0.137 |
| motorcycle | 0.278 | 0.191 | 0.333 |
| bicycle | 0.518 | 0.544 | 0.611 |
| mIoU | 0.515 | 0.476 | 0.581 |

**Table 1.** Class IoU and mIoU scores on Minicity test set for differently trained EfficientSeg architectures

## 5    Conclusions

In conclusion, we introduced a novel semantic segmentation architecture EfficientSeg, our architecture consists of scalable blocks which makes it easy to fit for problems of different scales. In our work we empirically show how selecting the most beneficial augmentation using the prior knowledge coming from the dataset improves the success of the network, making it even more advantageous than up-scaling the network. When trained with our augmentation set EfficientSeg (1.5) achieves 51.5% mIoU, outperforming its much larger counterpart EfficientSeg (6.0) if no augmentation is applied, in the other hand when trained with our augmentation set we achieve our best score 58.1%. Utilizing prior knowledge is especially important on tasks providing few data to train on, as the popularity of efficient image recognition networks increases, it is expected that data efficiency is the next step to have simple, efficient and elegant solutions to image recognition tasks.

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
