# OpenReview forum: "EfficientSeg: An Efficient Semantic Segmentation Network"
_thecvf.com/ECCV/2020/Workshop/VIPriors — Submitted to VIPriors_

### Official Review · AnonReviewer1 · 2020-07-21
**Effective method, out of scope**

**Confidence:** 4
**Rating:** 5

**Review:**

#### 1. [Summary] In 2-3 sentences, describe the key ideas, experiments, and their significance.
The paper proposes a novel CNN architecture for semantic segmentation based on U-Net and MobileNetV3 blocks. The proposed architecture is applied on the MiniCity dataset and performance improvements are shown.

#### 2. [Strengths] What are the strengths of the paper? Clearly explain why these aspects of the paper are valuable.
* Clarity: the paper is clear and easy to read.
* Effectiveness: the method seems effective on small-size datasets.

#### 3. [Weaknesses] What are the weaknesses of the paper? Clearly explain why these aspects of the paper are weak.
* Scope: The proposed architecture change is motivated by computational efficiency and parameter reduction rather than incorporating prior knowledge. Prior knowledge is only considered to motivate the techniques used for data augmentation, which are generic and widely used techniques.

#### 4. [Overall rating] Paper rating
* 5. Marginally below acceptance threshold

#### 5. [Justification of rating] Please explain how the strengths and weaknesses aforementioned were weighed in for the rating.
Despite being effective, the proposed architecture is not motivated by incorporating prior knowledge but rather by computational efficiency and therefore the paper falls outside of the intended scope of this workshop.

#### 6. [Detailed comments] Additional comments regarding the paper (e.g. typos or other possible improvements you would like to see for the camera-ready version of the paper, if any.)
(line 173) "We divide the learning rate by 10 at . "
Missing word.
(line 177) "poll" --> pole

---

### Official Review · AnonReviewer2 · 2020-07-27
**EfficientSeg: An Efficient Semantic Segmentation Network**

**Confidence:** 5
**Rating:** 4

**Review:**

#### 1. [Summary] In 2-3 sentences, describe the key ideas, experiments, and their significance.
The paper proposes new U-Net architecture which uses MobileNetV3 blocks.

#### 2. [Strengths] What are the strengths of the paper? Clearly explain why these aspects of the paper are valuable.
- New Unet


#### 3. [Weaknesses] What are the weaknesses of the paper? Clearly explain why these aspects of the paper are weak.
- Only changing the architecture and no prior rather than generic data augmentation
- It is more like technical report.
#### 4. [Overall rating] Paper rating
4

#### 5. [Justification of rating] Please explain how the strengths and weaknesses aforementioned were weighed in for the rating.


#### 6. [Detailed comments] Additional comments regarding the paper (e.g. typos or other possible improvements you would like to see for the camera-ready version of the paper, if any.)
- Why the 'related works' are related to the paper is missing.
- L.32: Is any meaningful pretraining available for medical data?
- Simple present tense instead of past tense
- Fig.1 image resolution
Typos:
- L.173: 10 at .
Missing citations:
- L.24: "..appoaches.." (only one is given.)
- L.26: "..various problems.."
- L.34: "..newly emerging..". What are those fields?
- L.57: classifying objects[?,?,?], detecting objects[?,?,?], estimating pose[?,?,?]
- L.72: easy-to-access applications[?,?,?]
- L.61: offline[?,?,?].., real-time[?,?,?]

---

### Decision · Program_Chairs · 2020-07-29

**Decision:**

Reject

**Comment:**

After considering the reviews and further discussion, we do not find sufficient cause to overturn the recommendation of the reviewers.